# MOEfication by Experts as Masks

## Abstract

In this work, we investigate how to sparsify a pre-trained dense large language model into a mixture-of-experts (MoE) architecture for faster inference. Our approach applies mask matrix to the activations for each expert, constrained by $L_0$ regularization to minimize the number of activated parameters. Starting with all parameters active, the model is progressively sparsified during training, ensuring minimal performance loss. This approach proves more efficient than one-shot sparsification techniques (Zhang et al., 2022), which typically require significant resources for performance recovery. Moreover, our approach automatically identifies shared, token-specific, and inactive experts, allowing for more efficient allocation of computational resources. Through extensive experiments, we achieve up to 97% performance retention on downstream tasks with only 50% of the feed-forward parameters activated in dense models. Beyond enhancing inference efficiency, this strategy of sharing computational units among experts presents a valuable framework for designing more generalized and efficient MoE architectures, opening avenues for future advancements in expert-based models.

## 1 Introduction

Under the guidance of scaling laws, the parameter count in large language models (LLMs) has continued to rise, with models ranging from LLaMA 7B to 70B parameters. To alleviate the substantial computational burden associated with model inference and deployment, various model compression techniques have been proposed. However, their application to LLMs often results in unacceptable degradation of performance. Thus, a critical challenge remains: *how to effectively reduce inference computation without compromising model efficacy?*

Sparse activation presents a promising solution. A notable example is the **Mixture-of-experts (MoE)** approach, which designs multiple expert structures with extensive parameters but activates only a subset during computation. This limits the number of active parameters and effectively mitigates the computational load. Despite the effectiveness of current sparse activation methods, they typically require training from scratch, which incurs prohibitive computational costs. An alternative research direction explores converting existing dense models into sparsely activated ones. Techniques such as MoEfication (Zhang et al., 2022), LLaMA-MoE (Zhu et al., 2024), and Turbo Sparse (Song et al., 2024) exemplify this approach by treating specific dimensions of the weights in the feed-forward network (FFN) as expert structures, selectively activating these dimensions during forward computation. Although these methods avoid the need to retrain from scratch, they rely on heuristic-based expert construction (*e.g.,* equally distributing weight dimensions across all experts), which neglects the varying significance of different dimensions within large language models. This can lead to suboptimal performance, as it overlooks the fact that some dimensions can be pruned while others can be shared across experts.

Table 1: Comparison of MoM and MoE. "Flexibility" refers to the adaptability in expert structure design, "Mem" indicates memory usage, and "Training Cost" reflects the computational budget required for training.

| Methods | Flexibility | Mem | Training Cost |
|---|---|---|---|
| MoE | ✗ | ✗ | High |
| MoEfication | ✗ | ✔ | Low |
| **MoM** | ✔ | ✔ | Minimal |

To address these challenges, our approach follows the principle of *maximizing efficiency while maintaining model performance and structure*. Specifically, inspired by MoEfication (Zhang et al., 2022),

we propose transforming the dense FFN structure into a sparse Mixture-of-Experts (MoE) module using a routing mechanism for selective activation of parameters. However, achieving activated sparsity with MoEfication style is non-trivial due to following practical challenges:

1. Construct experts by identifying the varying importance of different weight dimensions.

2. Minimize performance degradation during the conversion from a dense to a sparse model.

To achieve this, we develop a learning-based expert construction mechanism that dynamically assigns different dimensions to experts during the continue pre-training phase, based on the varying importance of dimensions. Furthermore, we propose an efficient training method that aims to maximize activation sparsity while minimizing performance degradation.

We propose a novel sparsification method for large language models, called *Mixture-of-Masks (MoM)*, which dynamically selects and activates a subset of parameters through learning-based masks. This approach reduces computational overhead while maintaining model performance, offering an efficient solution for balancing sparsity and effectiveness. MoM achieves expertization by integrating mask matrices into the FFN structure, where the mask vectors serve as substitutes for expert modules. These masks, composed of $\{0,1\}$ values, determine which dimensions to activate during training. Through this mechanism, we can: **(1) Adaptively learn which dimensions to share, token-specific, or prune**. By training the masks with $L_0$ norm constraints, we retain only the dimensions crucial for the current token, enabling automated expert construction without relying on heuristic-based methods, thereby eliminating prior biases. **(2) Perform lossless pruning for efficient continued pre-training**. We initialize the masks with all ones, ensuring that model performance remains unaffected during the initial pruning phase, allowing for the integration of multiple compression techniques.

We conducted comprehensive experiments to evaluate the performance of MoM, focusing on model accuracy restoration, data efficiency, and inference costs. In publicly available evaluation benchmarks, MoM outperformed existing methods with fixed expert allocation, restoring 97% of the dense model's accuracy compared to 90% achieved by MoEfication (Zhang et al., 2022). MoM effectively maintains model performance while exhibiting superior data efficiency during training. In addition, starting from the original dense model, MoM gradually prunes parameters with minimal accuracy loss, achieving the compression target after processing just 10B tokens. In contrast, methods with fixed expert allocation introduce significant structural changes, resulting in prolonged training times to restore model accuracy.

In addition, we also conducted an in-depth analysis to shed light on why MoM works well. Upon analyzing the experts obtained through MoM training, we observed that the experts were automatically divided into shared experts, independent experts, and ineffective experts. Both shared and ineffective experts can be excluded from routing, thereby reducing the model's inference costs and further improving efficiency. This observation is consistent with conclusions from some of the most advanced model structures, opening new directions for us to explore the characteristics of MoE architectures.

## 2 METHODS

In this section, we introduce Mixture-of-Masks (MoM), a novel sparsification method designed to produce compact models by selectively activating a subset of parameters. This approach achieves sparsity and computational efficiency while maintaining strong performance within a modest resource budget.

### 2.1 PRELIMINARY

We first present the background for our approach to mixture-of-experts architecture and the pruning methods.

**Mixture-of-Experts.** The MoE architecture enhances model capacity by increasing the number of parameters, but only activates a subset during computation, minimizing the computational cost. Typically, this involves duplicating the Feed-Forward Network (FFN) multiple times within the Transformer block, with only a subset of these "experts" active at any given moment. Inspired

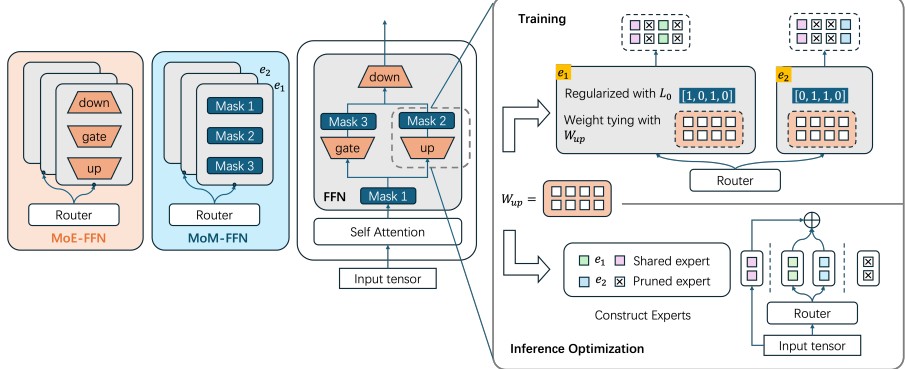

Figure 1: Overview of MoM architecture. MoM-FFN trains multiple masks as experts instead of multiple copies. For training, the masks are regularized by $L_0$ normalization. For inference, we construct experts with identified expert patterns.

by this, recent work has shown that transforming a dense model into an MoE structure effectively achieves activation sparsity. Formally, the output of MoE architecture $y$ can be computed as:

$$h = \Sigma_{i=1}^{n} G(x) \cdot E_i(x), \tag{1}$$

where $G(x)$ and $E_i(x)$ are the output vectors of the gating network and the $i$-th expert for a given input $x$, respectively. However, current methods randomly allocate dimensions, disregarding the varying importance of each dimension. This non-optimal allocation often results in performance degradation. Therefore, there is a pressing need for a method that can establish experts tailored to the model and pre-training data.

**Learning the Masks.** Pruning aims to achieve sparsity in large models by removing less important weights or components. Common approaches include structured pruning (removing specific structures) and unstructured pruning (removing individual weights). However, for Large Language Models, scaling laws indicate that a large number of parameters is crucial for optimal performance. Directly reducing the total number of parameters can harm the model's capacity. Therefore, we propose the concept of "activation pruning", which maintains the total number of parameters while pruning only the activated ones. This approach aims to preserve the model's advanced capabilities while reducing computational costs. In this context, we follow the study Louizos et al. (2017) of $L_0$ regularization to constrain the sparsity of large language models.

## 2.2 CONSTRUCTING EXPERTS BY MASKS

Following the work (Zhang et al., 2022), we treat the *dimensions* of weights in FFN as the minimal unit, and *experts* are constructed by grouping multiple dimensions together. Instead of manually assigning dimensions to experts, our objective is to dynamically group related dimensions into experts based on their interrelationships. In this section, we introduce **M**ixture-**o**f-**M**asks (MoM), a mask-based expert construction approach that enables dynamic selection of dimensions.

To implement this, we consider a LLaMA AI@Meta (2024) style decoder-only model with $N$ Transformer layers. Then the output of FFN can be described as follows:

$$h = F(\mathbf{W}^g x) \cdot \mathbf{W}^u x, \tag{2}$$

where $\mathbf{W}^g, \mathbf{W}^u \in \mathbb{R}^{e \times d}$ are the weight of gate and up projections and $F(\cdot)$ is the activation function. Our goal is to insert mask variables (denoted as $\mathbf{v} \in \mathbb{R}^d$) at various positions in this formulation to achieve sparse activation of different components. Depending on where the masks are inserted, we then introduce our method within two steps: (1) basic masking method by selecting expand intermediate dimensions in the FFN, and (2) fine-grained method with three strategies to further increase sparsity.

**Basic Masking Method.** The basic characteristic of the FFN structure is that expanding through the gate and up components can increase model capacity, but it also introduces significant redun-

dancy. Our approach involves adding a mask module with values $\{0,1\}$ after the gate and up outputs. Then, the output of the FFN becomes:

$$h = [F(\mathbf{W}^g x) \cdot \mathbf{W}^u x] \odot \mathbf{v}. \tag{3}$$

The masks are dynamically learned (see Section 2.3) rather than being statically assigned, as in previous work (Zhang et al., 2022; Zhu et al., 2024). This dynamic approach allows dimensions corresponding to similar tokens to be grouped together after training, aligning with the core idea of the MoE structure, *i.e.,* similar tokens activate similar sets of parameters, improving both efficiency and specialization. Finally, the sparsity calculation method is:

$$R(\mathbf{v}) = \frac{\sum_{i=1}^{d} \mathbb{I}(v_i = 0)}{d}, \tag{4}$$

where $\mathbb{I}$ is a indicator function.

**Fine-grained Masks Strategies.** While the basic masking method provides an initial reduction in redundancy, further improvements can be achieved by targeting specific components of the FFN with more fine-grained masking strategies. Because each component, such as the gate and up projections, may require different sparsity levels based on their relative contribution to the model overall performance, allowing for more granular control over the sparsity (Song et al., 2024).

Then, we extend the masking approach to fine-grained modules (*i.e.,* gate, up, and hidden states separately). For gate and up projections, the final sparsity is calculated as $R_{FFN} = (R_{gate} \odot R_{up})$. Additionally, we add masks to the FFN inputs, considering the differences between inputs where only a few dimensions need to be expanded to higher dimensions. This approach results in a sparsity calculation of $R_h \odot R_{FFN}$.

## 2.3 TRAINING WITH $L_0$ REGULARIZATION

Building on the mask construction strategy described earlier, the final set of experts is determined by the parts of the model that are retained by the learned masks. To increase sparsity and reduce the number of active parameters, we frame this as a constrained optimization problem. Here, the goal is to learn mask matrices that select sub-dimensions corresponding to specific tokens, while still maintaining overall model performance.

Inspired by the $L_0$ regularization method (Louizos et al., 2017), we parameterize the masks to model hard concrete distributions. These distributions are defined on the interval $[0, 1]$ but concentrate their probability mass at 0 or 1, enabling discrete decisions to either prune or retain specific dimensions. In addition, by starting with all parameters active, the model is progressively sparsified during training, ensuring minimal performance loss.

To formalize this process, let $l$, and $E$ represent the number of layers and the number of experts per layer, respectively. Given a target sparsity ratio $R_t$, the optimization objective for each layer is defined as:

$$L_{mask} = \sum^{l} \sum^{E} (R_e - R_t) + (R_e - R_t)^2, \tag{5}$$

where $R_e$ denotes the actual sparsity of the layer after mask application. Combined with the language modeling loss, the final loss is $L_{lm} + L_{mask}$. This formulation encourages the model to achieve the desired sparsity while minimizing the impact on performance.

Since each expert learns independently, the model naturally categorizes dimensions into three types: *shared dimensions* (across all experts), *independent dimensions* (specific to individual experts), and *unused dimensions* (not allocated to any expert). By automating this process, we reduce the risk of introducing prior biases and improve the efficiency of the model's sparse activation mechanism. Then we will introduce inference optimization based on identified expert types.

## 2.4 INFERENCE OPTIMIZATION VIA EXPERT PATTERN IDENTIFICATION

In this section, we optimize inference by leveraging the expert patterns identified through the $L_0$ regularization process. Specifically, we categorize experts into three groups: *shared experts*, *independent experts*, and *redundant experts*. This classification allows us to apply customized strategies for each type:

- **Shared experts.** Shared experts are dimensions that remain active across all experts. These are processed only once, as their outputs can be reused across different inputs, thereby reducing memory usage and computational load.

- **Independent experts.** For independent experts, we introduce a routing mechanism that selectively activates experts, following the standard MoE routing strategy. This approach helps to significantly decrease computational costs by activating only the necessary experts.

- **Redundant experts.** Redundant experts are dimensions that are never routed across any of the experts. These dimensions are pruned, as their contribution to model performance is negligible, further reducing the overall parameter count.

Interestingly, several advanced studies (Dai et al., 2024) have manually divided experts into shared and independent groups, arguing that shared experts capture common knowledge while independent experts focus on domain-specific tasks. Our findings after applying MoM are consistent with this, but in our case, the model automatically learns this division. To provide deeper insights into the underlying rationale, we conduct a more detailed analysis of the expert patterns, which is presented in Section 3.4. This analysis sheds light on the architectural design principles that guide the optimal allocation of experts.

## 2.5 DISCUSSION

A closely related approach to the $L_0$ regularization-based sparsification presented in this paper is pruning, which directly compresses the total number of parameters. For example, Sheared LLaMA (Xia et al., 2024) removes unimportant structures, while SparseGPT (Frantar & Alistarh, 2023) masks redundant values in the weight matrices. However, reducing the total number of parameters can limit the model capacity to capture complex patterns, which contradicts the goals of large language models.

In contrast, our method focuses on selective activation of parameters, preserving the model's full capacity while significantly reducing computational overhead. Through our experiments, we found that this approach is more suitable for large models compared to direct pruning. Specifically, we observe two key advantages: (1) *Balancing model capacity and computational efficiency*: Selective parameter activation requires fewer data to recover model performance compared to total parameter pruning. (2) *Scalability for larger models*: In our experiments with LLaMA-3-8B, we find that achieving a compression rate of 50% required only 20B tokens.

## 3 EXPERIMENTS

In this section, we first set up the experiments and then report the results and analysis. Then we conduct a detailed analysis under different MoE settings.

### 3.1 EXPERIMENTAL SETUP

**Datasets.** For continue pre-training process, we aim to restore the performance when selectively activating a subset of the parameters. So we use a mixture of several data sources to cover several domains, including: (1) RedPajama (Computer, 2023), a mixture of CommonCrawl, C4, Github, Wikipedia, Books, arXiv, and StackExchange. We try to cover a diverse set of domains for a better performance restoration. (2) Dolma (Soldaini et al., 2024), built from a diverse mixture of web content, scientific papers, code, public-domain books, social media, and encyclopedic materials. (3) SkyPile (Wei et al., 2023), a large-scale Chinese dataset containing approximately 150B tokens. For evaluation, we follow the study (Wei et al., 2023; Zhu et al., 2024) and utilize HellaSwag to evaluate the model ability since the performance on HellaSwag is reported to grow smoothly during pre-training.

For a comprehensive assessment of downstream tasks, we follow Sheared LLaMA (Xia et al., 2024) and use lm-evaluation-harness package (Gao et al., 2024) to evaluate the following tasks: BoolQ (Clark et al., 2019), PIQA (Bisk et al., 2020), SiQA (Welbl et al., 2017), HellaSwag (Zellers et al., 2019) and ARC easy (Clark et al., 2018).

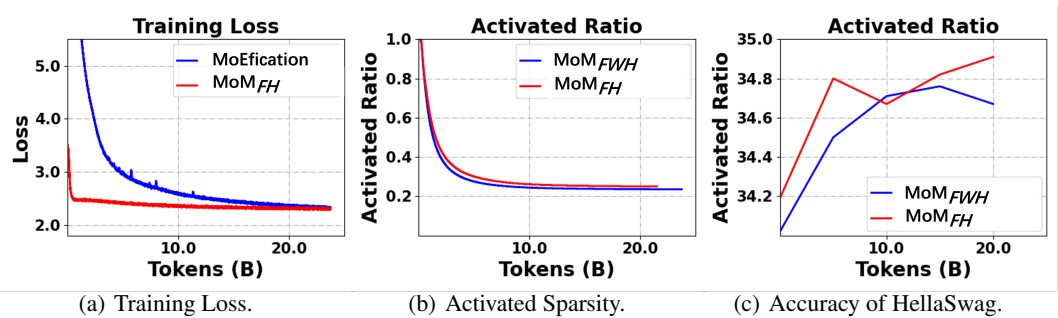

(a) Training Loss.  (b) Activated Sparsity.  (c) Accuracy of HellaSwag.

Figure 2: Model loss and activated sparsity. (a) shows the comparison between MoM and MoEfication. (b) and (c) illustrate the compression rate and downstream task performance of our method under the fine-grained masking strategy.

**Implementation.** For the implementation of continued pre-training setting, we utilize the open-source SkyWork model (Wei et al., 2023) with 300M parameters for our experiments. SkyWork provides a general LLaMA-style model framework, ensuring that our method can be easily transferred to other similar frameworks. Additionally, since all data associated with this model is accessible, it provides a fair platform for comparing the effectiveness of different methods. Based on this model, we start from a checkpoint trained with 200 billion tokens. According to the Section 2.2, we provide four variants of different masking strategies: MoM, $\text{MoM}_{FH}$, $\text{MoM}_{FW}$ and $\text{MoM}_{FWH}$:

• **MoM** is the base variant that only masks the intermediate dimensions in FFN module.

• **$\text{MoM}_{FH}$** use fine-grained masks of the hidden states dimensions based on MoM.

• **$\text{MoM}_{FW}$** use fine-grained masks to weights (*i.e., gate projection* and *up projection* separatedly).

• **$\text{MoM}_{FWH}$** is a combination of $\text{MoM}_{FH}$ and $\text{MoM}_{FW}$ to achieve higher sparsity ratio.

Subsequently, we assess the efficacy of various methods in restoring model performance under constrained training resources. To further demonstrate the scalability of our approach, we also conduct experiments on a larger LLaMA-3-8B model (AI@Meta, 2024). In the next section, we will present the detailed experimental results.

**Baseline Models.** Here we introduce relevant methods as our baselines.

• **MoEfication** (Zhang et al., 2022) for sparse activation. MoEfication converts dense models into a MoE version by splitting the FFN weights into multiple partitions as experts, with dimensions evenly distributed across experts.

• **Pruning.** We additionally employ model pruning as a baseline to validate the effectiveness of activation-based compression in comparison to full parameter pruning. Specifically, when the total number of experts is set to 1, our method reduces to traditional pruning, effectively compressing the total number of parameters. We use this configuration as a variant of pruning to provide a comparative baseline.

### 3.2 MAIN RESULTS

**Comparing with MoEfication.** First, we show dense downstream task evaluation results on both dense models and activated pruning methods. As shown in Table 2, MoM uses limited training tokens and outperforms MoEfication in all tasks. Specifically, MoM preserves 98% of original dense model (49.1 *vs.* 50.3), while MoEficaiton only preserves around 90% (45.1 *vs.* 50.3).

As for the data efficiency, we observe obviously from Figure 2 (a), that our method (red curve) converges to the same loss as the MoEfication (blue curve) very quickly, whereas MoEfication requires approximately 20B tokens to achieve a similar loss. This result indicates that using a lossless compression method in MoM can effectively enhance the data utilization efficiency than one-short sparsification like MoEfication.

Table 2: Models with MoM outperforms publicly available methods of sparsification. Models with "†" are our reproduced result.

| Model (#tokens for training) | #Activated | Commonsense & Reading Comprehension | | | | | Average |
| | | BoolQ | PIQA | SiQA | HellaSwag (10) | ARC-E | |
|---|---|---|---|---|---|---|---|
| Dense (200B) | 100% | 58.4 | 67.8 | 39.1 | 36.9 | 49.5 | 50.3 |
| MoEfication (20B)† | 50% | 59.4 | 58.5 | 36.5 | 29.3 | 42.0 | 45.1 |
| MoM (20B) | 75% | 60.0 | 66.9 | 36.3 | 35.3 | 46.6 | 49.0 |
| MoM$_{FH}$ (20B) | 50% | 59.5 | 65.6 | 37.2 | 34.9 | 48.2 | 49.1 |

As for the effect of our method during the compression process, Figure 2 (b,c) shows that the recovery of model performance remains stable across various compression rates. Specifically, performance recovery stays within 92% of the dense model (*i.e.,* 34.2 *vs.* 36.9), indicating minimal degradation even with significant compression. In the early stages of training, there is a slight drop in performance, despite a low loss value, but this is quickly corrected as training continues. The overall trend suggests that our method ensures performance stabilizes and recovers effectively. These results confirm the robustness of our approach, demonstrating that it achieves substantial compression without severely affecting model accuracy.

To demonstrate the scaling effect, we extend to the LLaMA3-8B model (see Figure 3). As for data preparation, existing work has shown that more complex datasets are often required to recover the model after compression, including data ratios (Xia et al., 2024) and larger data sizes (Zhu et al., 2024). Therefore, we adopt a classic dataset preparation pipeline to ensure a fair comparison. The results show that our method can still achieve faster model compression on the 8B model. It is worth noting that in LLaMA-8B, the compression process can be completed more quickly, requiring only a budget of 15B tokens. However, model recovery is a more prolonged process. Overall, the model performance gradually improves, while the recovery process for MoEfication might be a more long-term task. This demonstrates that MoM offers greater data efficiency compared to MoEfication.

**Comparing with Pruning.** To highlight the advantages of reducing activated parameters over pruning the total number of parameters, we constrain the number of experts to 1, effectively simulating a pruning-based approach, and compare the results with MoM$_{FH}$. The outcomes are presented in Figure 4 (a,d). Our findings indicate that the pruning method struggles to achieve lower compression rates, likely due to the challenge of balancing model performance and compression. As the compression

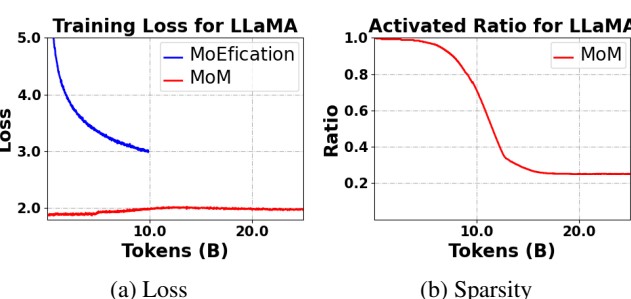

(a) Loss  (b) Sparsity

Figure 3: Extending experiments on LLaMA-3-8B.

rate decreases, maintaining model performance becomes increasingly difficult. In contrast, MoM$_{FH}$ easily achieves higher compression rates while preserving performance, demonstrating that activation sparsity is a more effective strategy for performance efficiency, particularly in large models.

### 3.3 DETAIL ANALYSIS

Here we provide detailed studies of two important aspects of learning masks: masking strategies and learning strategies.

**Masking Strategies.** In this section, we investigate different settings of mask strategies, including **(1) MoM$_{FH}$** remove dimensions in the input hidden states, **(2) MoM$_{FW}$** only remove the weights of the gate and up projections, and **(3) MoM$_{FWH}$** additionally remove weights in the gate

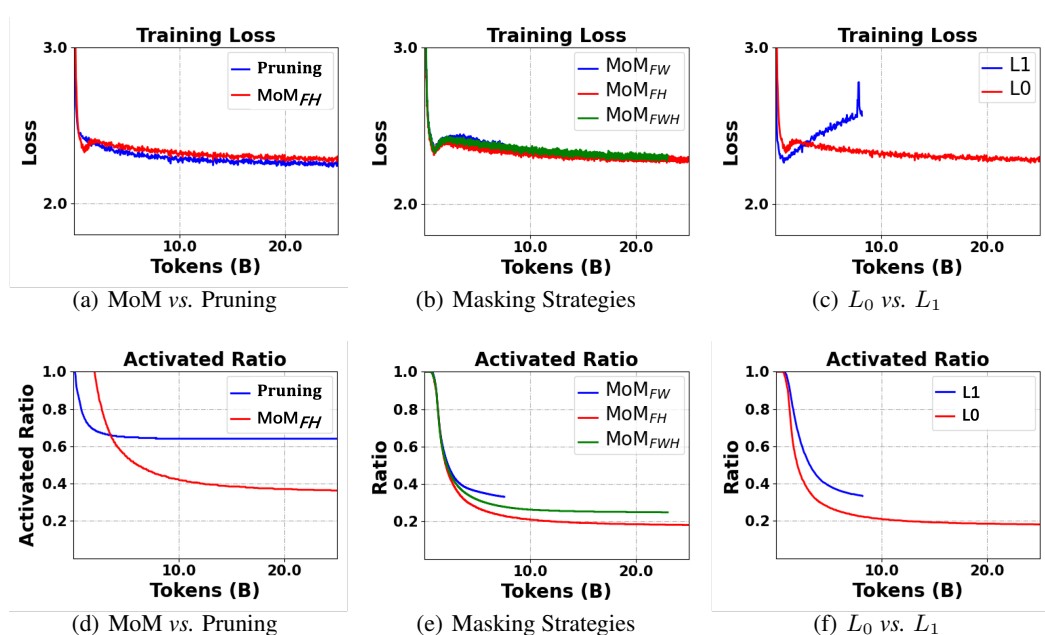

Figure 4: Influence of Masking Strategies for different metrics. Figures (a,d) denote the comparison with pruning. Figures (b,e) denote the ablation of different masking strategies. Figures (c,f) denote the ablation study of different learning strategies.

and up projections based on hidden states. Then we continue pre-train the 300M models with 20B tokens and report the evaluations on Hellaswag datasets in the Figure 4 (b,e). From the sparsity ratio, we find that $MoM_{FH}$ achieves a lower sparsity ratio than the others. Meanwhile, these compression gains sacrifice the performance as we can see from the evaluation in Hellaswag. For the 300M model, we find that $MoM_{FH}$ consistently performed the best. Therefore, we recommend prioritizing $MoM_{FH}$ for initial trials. However, if a larger training budget is available, $MoM_{FWH}$ may be more advantageous as it may lead to more sparsified models.

**Learning strategies.** In practice, optimizing binary masks can be challenging due to their discrete nature. Therefore, it is crucial to design an appropriate technique for learning effective masks. Popular approaches include normalization methods such as $L_1$ and $L_0$ regularization. To evaluate the effectiveness of these techniques, we performed an ablation study and present the results in Figure 4 (c, f). As shown in the figure, applying $L_1$ regularization results in a significant degradation in model performance at the early stages of training, with the loss rapidly increasing. This indicates that $L_1$ is not well-suited for sparsification tasks. Consequently, we halted the $L_1$ experiment after training with less than 10B tokens, as the sparsity achieved was considerably lower compared to $L_0$. In contrast, the $L_0$ regularization technique proved to be much more effective in achieving sparsity, validating its suitability for tasks involving sparse activation.

### 3.4 ANALYSIS FOR THE EXPERTS

**Experts Selection Across Layers.** In the Section 2.4, we propose that different experts, represented by individual dimensions, should have varying levels of significance in the model. Our method uses an adaptive training approach to assign dimensions into three categories: *shared experts*, *independent experts*, and *redundant experts*. By distinguishing the roles of each expert, the model can better allocate importance, improving both efficiency and interoperability.

To further understand this result, we visualize the experts at different layers, as shown in the Figure 5. We observe varying levels of preference for the experts across layers. For example, Expert 2 shows a relatively even level of participation, with activation remaining below 50% and spread across all layers. In contrast, Expert 4 exhibits activation in some layers that reaches approximately 80%,

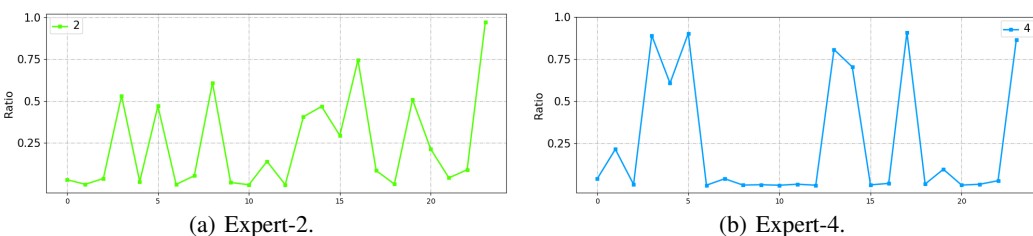

(a) Expert-2.          (b) Expert-4.

Figure 5: Visulizaton of experts selection.

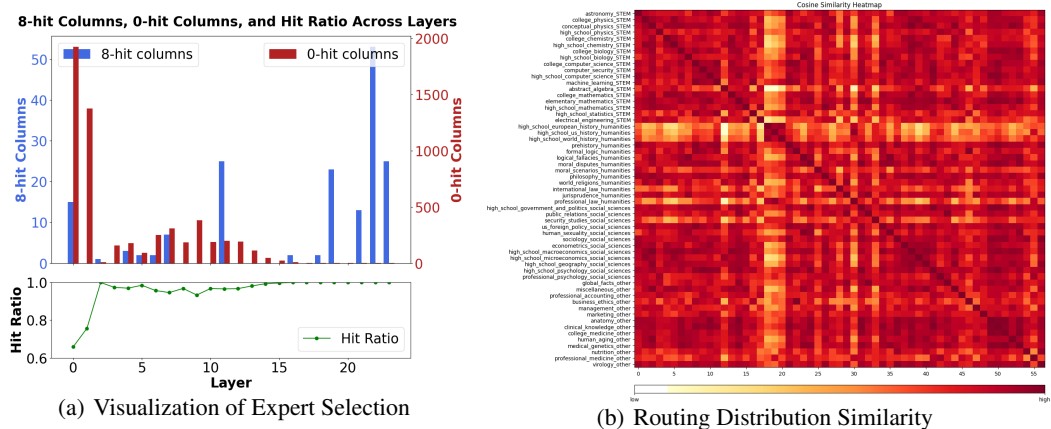

(a) Visualization of Expert Selection     (b) Routing Distribution Similarity

Figure 6: Analysis of the experts. (a) denotes the visulizaton of experts selection and (b) denotes the routing distribution similarity across MMLU 57 tasks.

but the number of activated layers remains relatively low, around 30%, which maintains higher efficiency.

Then we analyze the roles of shared, independent, and redundant experts across layers and their relationship to activation sparsity. Specifically, we use 8-hit dimensions to represent shared experts (blue bars) and 0-hit dimensions to represent redundant experts (red bars), see Figure 6 (a). Our analysis reveals two intriguing patterns: **(1) in shallow layers, more experts are redundant, and the model focuses on common, token-agnostic information.** This leads to higher activation sparsity, with many parameters deemed unnecessary. As we move to deeper layers, sparsity decreases, suggesting that the model requires more experts to handle the increasing complexity of semantic information. **(2) In the deepest layers (21, 22, and 23), we observe a rise in shared experts, even though these layers handle more complex and nuanced semantic tasks.** This implies that, despite the increased task complexity, there are underlying patterns or features that remain consistent across tasks, captured effectively by shared experts. This discovery points to the model's ability to extract cross-task or cross-domain information, a feature that may contribute to its generalization capabilities. Our findings offer valuable insights into the interpretability and efficiency of deep MoE models, showing how expert roles evolve across layers. Understanding these dynamics could lead to more efficient model architectures that balance the trade-off between task-specific adaptations and shared knowledge extraction.

**Experts Selection Across Tasks**    Then we empirically investigate whether different experts contain domain-specific information. For the dataset, we use the benchmark of MMLU where the tasks are categories into four groups (Hendrycks et al., 2021). First, we collect the output of the gate projections across all the layers and form a gate distribution vector of the dimension of 8 (experts per layer) × 24 (layers). Then we calculate the cosine similarity of the vectors and report the results in the Figure 6 (b). We observe a clear boundary between the `STEM` and `humanities` subjects, as shown by the clustering patterns in the heatmap. Additionally, three history tasks—`high`

`school european history,` `high school US history,` and `high school world history`—exhibit strong correlations with each other, more so than with other tasks. This is likely due to the significant overlap in the subject matter across these history topics, which makes them more similar compared to other tasks.

Notably, even though our experts are constructed using masks rather than the traditional MoE approach, they still successfully learn to capture domain-specific information and categorize tokens based on their content. This demonstrates that our approach retains the essential characteristics of traditional MoE models while offering greater flexibility.

## 4 RELATED WORK

**Pruning.** Existing models are often impractical to deploy due to their large parameter count. A direct solution to this issue is pruning (Xia et al., 2024), which involves the removal of model weights. Pruning generally follows two primary approaches. The first approach is structured pruning (Xia et al., 2024), which typically achieves higher compression rates and enhances inference efficiency. However, this method often results in significant performance degradation due to the coarse granularity of pruning, which inadequately preserves essential weights. Consequently, extensive retraining is often necessary to recover model performance. The second approach is unstructured pruning (Song et al., 2024; Wang et al., 2024), which eliminates non-essential weight values. This finer-grained method effectively retains important weights, resulting in minimal performance loss. However, it does not substantially improve inference speed. The traditional work focus on reducing the total parameters which may not against the spirit of scaline law (Kaplan et al., 2020): the large language models where the superior ability comes from a large number of parameters.

**Sparsed Methods.** In contrast to pruning, activating fewer parameters during computation maintains model capabilities without increasing computational load, making it an effective augmentation strategy. A typical approach is the Mixture of Experts (MoE) structure (Fedus et al., 2022; Lepikhin et al., 2020), where multiple FFN structures act as experts, with only a subset activated for computation, effectively reducing parameter count. Numerous studies have validated the efficiency of this method in large-scale models. For instance, the Mixtral (Jiang et al., 2024) model implements a standard MoE structure at a 7B scale, while DeepSeek (Dai et al., 2024) enhances MoE by incorporating shared experts for common knowledge and unique experts for specific tasks. Additionally, existing pre-trained models can be transformed into MoE structures by replicating the FFN multiple times and activating only a few each time. This process, termed "MoEfication" (Zhang et al., 2022) has successfully modified smaller models like BERT and larger ones like Llama-MoE (Zhu et al., 2024). Although these methods effectively leverage the knowledge of existing models, the structural changes often lead to performance degradation. This paper focuses on enhancing the effectiveness of MoEfication to establish it as a viable solution.

## 5 CONCLUSION

We introduced Mixture-of-Masks (MoM), a novel method to transform an existing dense model into a sparsely activated architecture, offering high efficiency while maintaining performance. With MoM, we achieved 97% of the performance of the dense counterpart, with only 50% of the feed-forward network (FFN) parameters activated, significantly reducing computational costs under a 10B parameter training budget. Compared to the traditional Mixture-of-Experts (MoE) approach, MoM had been demonstrated superior efficiency in both parameter usage and computation. In addition to its performance gains, we also provided valuable insights into the distribution of experts, revealing key design principles that can inform the construction of more interpretable and efficient MoE architectures. These findings not only improve our understanding of how to optimize sparse models but also suggest new directions for enhancing the balance between performance and efficiency in large-scale language models. For future work, we plan to extend our method to more components within the model architecture, including attention weights and even embeddings. we aim to further improve the model's parameter efficiency and achieve greater computational savings.

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
