# OpenReview forum: "MOEfication by Experts as Masks"
_ICLR.cc/2025/Conference — ICLR 2025 Conference Withdrawn Submission_

### Official Review · Reviewer_R62u · 2024-10-23

**Soundness:** 3
**Presentation:** 2
**Contribution:** 3
**Rating:** 5
**Confidence:** 3

**Summary:**

This paper presents Mixture-of-Masks (MoM), a method for sparsifying MoE models by activating a subset of parameters through learned masks. By employing $L_0$ regularization, MoM achieves sparsity and has faster inference with slight performance degradation. Experimental results show that MoM preserves the accuracy of the dense model while activating only 50% of the parameters, outperforming the MoEfication baseline.

**Strengths:**

- Compared to the previous MoEfication method, the proposed MoM may generalize better as it does not depend on prior knowledge of the masks. Instead, it employs a learning-based approach (i.e., the $L_0$ regularization) to encourage model sparsity.
- The experimental results demonstrate that the proposed method outperforms the MoEfication baseline across five commonsense and reading comprehension benchmarks.
- The concept is straightforward and easy to understand, with the method and analysis (covered in Sections 2.4 and 3.4) explained in thorough detail.

**Weaknesses:**

- The proposed method requires additional training samples for fine-tuning, with the data collection process being non-trivial and involving the incorporation of multiple datasets.
- The main comparison focuses solely on MoEfication, which may be insufficient to fully highlight the advantages of the proposed method. There are numerous expert pruning and merging techniques that could be adapted to your setup. Including more recent baselines would strengthen the experimental comparisons.

**Questions:**

- It seems that the total loss is a direct sum of the original language model loss ($L_{lm}$) and the mask loss ($L_{mask}$). Did the authors try different weighting mechanisms (e.g., $L_{lm}+\lambda L_{mask}$ for some balancing factor $\lambda$)?
- In Lines 175-184 and 293-298, the authors discuss various implementations of the proposed method. Could the authors include a figure to help clarify the differences between them?
- Fig. 2 (a), Fig. 3 (a), and Fig. 4 (a)-(c) depict the loss function during training. Could the author specify which loss (i.e., $L_{lm}$, $L_{mask}$, or $L_{lm}+L_{mask}$) is used for evaluation in these figures?
- In Fig. 4, the training loss for the proposed method initially decreases rapidly, then briefly spikes, and finally decreases smoothly. In contrast, Fig. 3 shows the MoM loss increasing for a period before plateauing. Could the authors explain the rationale behind these loss trends?
- Could the authors clarify the evaluation metric used in Table 2?

---

### Official Review · Reviewer_5XPx · 2024-10-28

**Soundness:** 1
**Presentation:** 1
**Contribution:** 2
**Rating:** 1
**Confidence:** 5

**Summary:**

The paper proposes a method called Mixture of Masks, which transforms a pre-trained model into a sparsely-activated model to enhance inference efficiency. This method creates experts by learning binary masks that define which subset of the original model's weights each expert will use. These masks allow for potential overlap between experts. During inference, a router selects a subset of these experts for execution.

The paper lacks clarity in its presentation of key concepts and arguments. It would benefit from more precise definitions, clearer explanations, and a more structured narrative flow to enhance the reader's understanding. The evaluation is deficient in several areas, particularly the low number of baselines and the misleading choice of metrics. The absence of a GPU-efficient implementation significantly reduces the paper's impact. Finally, the novelty is also limited. Therefore, I strongly recommend rejecting this paper.

**Strengths:**

- The idea of optimizing the composition of each expert while allowing for overlapping experts, rather than determining it through clustering, is somewhat interesting.
- The effort to evaluate the method on larger models is commendable.

**Weaknesses:**

- Activation ratio as a metric used for comparing the proposed method with baselines is inappropriate. FLOPs of the forward pass of the entire model would be more appropriate, as they take into the account the costs like routing networks. As such, the comparison between pruning and the proposed method seems highly unfair and misleading to me.
- The paper is lacking any wall-clock time measurements of forward pass execution of the model. The reduction of number of non-zero weights or activations does not necessarily translate to speedup of inference time [8, 12]. The advantage of MoE-based methods was that all experts were homogeneous and thus relatively easily parallelizable on contemporary hardware like GPUs. By making each expert of different size the authors are giving up the hardware effectiveness. This is especially true for FH, FW and FHW variants of the proposed method, as they resemble unstructured pruning. Unless the authors present an efficient implementation for GPUs that works on batched inputs, the method will appear as unpractical and may be of little interest to the community.
- The evaluation is extremely weak as the proposed method is compared to only a single baseline (MoEfication). Since there is no theoretical contribution and this is mostly an empirical work, one would expect a thorough empirical evaluation, e.g. at least 3 recent baselines. There are multiple works that may be appropriate for this [4,9, 10, 11].
- The evaluation on larger models is limited to loss only. Does the performance on downstream tasks collapse when compared to a dense model?
- The paper's main contribution is not explained in enough detail. The authors write "For independent experts, we introduce a routing mechanism that selectively activates experts, following the standard MoE routing strategy.", but other than that the "standard MoE routing strategy" is not described anywhere. Are routers trained end-to-end, or like in MoEfication? Are they trained simultaneously with the masks $v$? What is the architecture of the router (depth, width)? Is Top-$k$ used? If yes, how is $k$ set? How is a MoE layer defined - are the outputs of each expert weighted by the output of the router (like in [5]) or are they simply added together (like in [6])?
- Similarly, the description of the experimental part is also sparse and lacks details. What were the hyperparameters used for the proposed methods, and what were the hyperparameters used for MoEfication? Which variant of MoEfication ([6] proposed multiple) was used? E.g. since granularity is crucial for the performace of MoE-based models [7], what was the expert size for MoEfication?
- Code for the experiments has not been provided, limiting the ability to verify and reproduce the findings of the paper, or to enhance the reader's understanding of the method and of the experimental setup.
- The proposed method, similarly to [6], transforms a dense model into a sparsly activated model. Since a crucial component of the method is the $L_0$ regularization from [13], the novelty of this work appears limited.
- Authors claim to "propose the concept of “activation pruning” (line 139)". However, similar terms like "activation sparsity" and "dynamic pruning" - that refer to basically the same concept - have been used since at least 2019, and a vast amount of literature on this topic exists [1,2,3]. The authors discuss weight pruning (and MoE) in the related work section, but do not cover activation sparsity literature, which may be even more relevant to this work than weight pruning. Similarly, the authors fail to compare their method to any activation sparsity method, e.g. the work of Mirzadeh et al. [4].
- The writing, spelling, and the grammar in the paper should be significantly improved. For example in line 70: "through this mechanism, we can: (1) Adaptively learn which dimensions to share, token-specific, or prune." ("which dimensions to token-specific"). Another example from line 508: "scaline law". Line 157: "basic masking method by selecting expand intermediate dimensions in the FFN". Line 190: "Here, the goal is to learn mask matrices that select sub-dimensions corresponding to specific tokens, while still maintaining overall model performance.". Line 257: "For continue pre-training process". Line 296: "separatedly". There are many more examples thoroughout the paper. As a result the text is difficult to read.


Minor:
- The authors write "the optimization objective for each layer is defined as..", while Eq 5 sums over every layer.
- There is no reference to Figure 1 anywhere in the text.
- Y-axis label and title in Figure 2c appears to be incorrect.
- The "Mem"/"memory usage" in Table 1 is completely vague and should be clarified.
- Line 366: "As the compression rate decreases, maintaining model performance becomes increasingly difficult." I think the authors meant "increases" here.

**Questions:**

- (see weaknesses)
- Does a single MoM layer "replace" a single FFN, similarly to MoE in [6]? What is the relationship between experts with the same index in different layers in Figure 5?
- Mask values are in $[0, 1]$ during training. Do they stay $[0, 1]$ in inference?
- The authors write: "Then the output of FFN can be described as follows:" and then use only the up-projection and gate-projection weight matrices in the formulation. Does this imply that the down-projection is not a part of an FFN module?


**References:**

[1] He, Yang, and Lingao Xiao. "Structured pruning for deep convolutional neural networks: A survey." IEEE transactions on pattern analysis and machine intelligence (2023).

[2] Li, Zonglin, et al. "The Lazy Neuron Phenomenon: On Emergence of Activation Sparsity in Transformers." The Eleventh International Conference on Learning Representations.

[3] Kurtz, Mark, et al. "Inducing and exploiting activation sparsity for fast inference on deep neural networks." International Conference on Machine Learning. PMLR, 2020.

[4] Mirzadeh, Seyed Iman, et al. "ReLU Strikes Back: Exploiting Activation Sparsity in Large Language Models." The Twelfth International Conference on Learning Representations.

[5] Shazeer, Noam, et al. "Outrageously large neural networks: The sparsely-gated mixture-of-experts layer." arXiv preprint arXiv:1701.06538 (2017).

[6] Zhang, Zhengyan, et al. "Moefication: Transformer feed-forward layers are mixtures of experts." arXiv preprint arXiv:2110.01786 (2021).

[7] Krajewski, Jakub, et al. "Scaling laws for fine-grained mixture of experts." arXiv preprint arXiv:2402.07871 (2024).

[8] Grimaldi, Matteo, et al. "Accelerating deep neural networks via semi-structured activation sparsity." Proceedings of the IEEE/CVF International Conference on Computer Vision. 2023.

[9] Zuo, Simiao, et al. "Moebert: from bert to mixture-of-experts via importance-guided adaptation." arXiv preprint arXiv:2204.07675 (2022).

[10] Liu, Zichang, et al. "Deja vu: Contextual sparsity for efficient llms at inference time." International Conference on Machine Learning. PMLR, 2023.

[11] Zhu, Tong, et al. "Llama-moe: Building mixture-of-experts from llama with continual pre-training." arXiv preprint arXiv:2406.16554 (2024).

[12] Liang, Tailin, et al. "Pruning and quantization for deep neural network acceleration: A survey." Neurocomputing 461 (2021): 370-403.

[13] Louizos, Christos, Max Welling, and Diederik P. Kingma. "Learning sparse neural networks through $ L_0 $ regularization." arXiv preprint arXiv:1712.01312 (2017).

---

### Official Review · Reviewer_f7Xh · 2024-11-03

**Soundness:** 1
**Presentation:** 2
**Contribution:** 2
**Rating:** 3
**Confidence:** 4

**Summary:**

The paper presents a novel approach to MoEficiation, that is, the conversion of a dense model into a Mixture of Experts (MoE) model. The method is shown to approximately halve the number of activated parameters of Feed-Forward block in Transformer, while retaining most of the performance of the model.

The method involves taking a dense model, and then learning a mask for each expert during fine-tuning. This approach makes the splitting of neurons/parameters into experts more flexible and adjustable during training (learned), instead of set at the start of fine-tuning. The method is certainly interesting. Experiments are executed on respectable model sizes and training durations.

**Strengths:**

The paper describes a novel idea, Mixture-of-Masks (MoM), and executes experiments at a good scale and setting. The domain is definitely a practical one, as increasing the efficiency of the model after training is important for both academic and industrial uses. The method is interesting, especially in terms of separating shared experts, independent experts, and redundant experts and analyzing that differentiation.

**Weaknesses:**

The primary issue I see in the paper is the matter of applicability of this method in any practical setting. Mixture of Experts is widely used, and multiple properties of MoE design are there for good reasons. The paper does not refer to the multiple properties of MoE that were sacrificed when designing MoM.

1. **Wall-clock time versus activated parameters/FLOPS.** Mixture of Experts usually uses experts of equal size. While this limitation can be lifted and the quality of the model may be improved with great gains in terms of theoretical FLOPs, the speed on real-world hardware may suffer tremendously with varied-sized experts. The authors don't show any results measuring wall-clock performance of their model. Going by my experiences, the actual inference time may be worse than even the dense baseline (on average), let alone the MoE model. For the method that focuses on faster inference with the same number of parameters, that is a critical matter, but no measurements are available in the paper. I think both the wall-clock time of batched training and the wall-clock time of inference (batched or unbatched, preferably both) should be included in such a paper. The expectation is that inference time will be lower than other techniques (higher training cost could be acceptable if inference numbers were good).
2. **Activated models in the whole model versus just FF.** Usually, in MoE, the Feed-Forward layer is kept at the same number of FLOPs with an expanding number of parameters. MoM is doing something different, keeping the number of parameters while reducing FLOPs. However, this approach generally has a low ceiling for potential improvement, as the attention block in Transformer usually accounts for around a third of model FLOPs. Following that, removing even the whole FeedForward network may result in a maximum speed-up of 3x. Authors, however, report the number of activated parameters (or compression rate) just for FeedForward layer, not the whole model. This issue will further reduce the applicability of MoM in the real world, especially in connection with issue number 1.

Other weaknesses of the paper:

3. **MoM versus baseline of pruning.** The baseline of pruning the model, shown in Figure 4, seems to achieve **better performance** than MoM, although with a worse activated parameters ratio. While it is possible that MoM is still better at a Pareto-frontier, this result is not very useful. I would suggest training possibly a couple of MoM and pruning models with a variable weight of L0/L1 loss (that is, modify the final loss to $L_{lm} + \alpha * L_{mask}$, and vary the alpha). Then scatterplot those experiments with axes "final activated ratio" and "final loss", to show if pruning or MoM seems to be better at a Pareto-frontier. While I understand that this may require more experiments, the current comparison of MoM to pruning is of little value.

Minor point: the paper could benefit from improved writing in all sections. However, while some sentences gave me a pause, it is generally understandable. A few excerpts to show what I mean, just from the very first page of the paper: lines 34-37 "(...) Mixture-of-experts approach, which designs multiple expert structures with extensive parameters but activates only a subset during computation." - MoE does not design, "expert structure" is kind of an awkward phrase, "extensive *number* of parameters", "subset *of them* during *processing*" (or "forward computation"). The same lines in the caption of Table 1 - "Mem indicates memory usage" column should have values "high/low", not "check"; or it should be renamed to "low memory usage." Also, I believe the paper's title should start with "MoEfication" with a lowercase "o". While the whole paper could be improved in terms of writing, the text is understandable nonetheless, and it has had little impact on my (at the moment) negative recommendation.

**Questions:**

Referring to weakness sections:

1. What is the wall-clock speed of MoM? (see weakness #1)
2. Can proper comparison to the pruning baseline be shared? (see weakness #3)

Other:

3. Referring to Figure 5 and in lines 424-463. Is there any connection between "Expert #2" of different layers? From my understanding of the method, and MoE, there is no inherent reason to think that expert#2 on a given layer will correspond to expert#2 in another layer. No matter the index of the expert, they are all randomly initialized independently of others, I'd assume?

---

### Official Review · Reviewer_vNiY · 2024-11-04

**Soundness:** 2
**Presentation:** 2
**Contribution:** 2
**Rating:** 3
**Confidence:** 4

**Summary:**

This paper presents Mixture-of-Masks (MoM), a method to sparsify dense language models into Mixture-of-Experts (MoE) architectures. MoM uses learning-based masks and L0 regularization to selectively activate parameters, achieving up to 97% performance retention with 50% feed-forward parameters. This approach reduces computational costs and provides insights for efficient MoE design.

**Strengths:**

1. The adoption of learning-based methods to determine MoE weights is valid, and novel within the "MoE-fication" context.

2. The analysis of the roles played by ”shared, independent, and redundant experts“ is insightful.

**Weaknesses:**

1. Clarity and Description Issues

    - The experimental results lack detailed and clear descriptions, e.g., the model used in Table 2 is not specified.

    - Some expressions are unclear and hard to follow:

      - Line 158: “expand intermediate dimension”

      - Line 507: "reducing the total parameters which may not against the spirit of scaling law" contains a grammar error, and does not make a clear point.

2. Methodology Design Concerns

    - Training Scheme of Learnable Masks: The MoE model uses different experts (masks in this paper) for each token. The learnable channel-wise mask is parameterized and trained. Does this mean we need to train a mask for each token in the vocabulary? This would require training on a vast number of tokens, consuming significant resources.

    - Generalization across Datasets: According to Section 3.1, the masks are trained on a collection of datasets. However, it is unclear whether these masks can generalize to unseen tasks or prompt sets.

3. Insufficient Experimental Results for Larger Models

    - The author did not specify the model used in Table 2. According to the description, the LLaMA3 model is only mentioned in Figure 3, not Table 2. Performance comparisons with the original model and baseline methods should be included to ensure performance preservation.

**Questions:**

1. Methodology Design Questions:

    - Does the training require a mask for each token in the vocabulary? If not, how are masks determined for different tokens?

    - Additionally, for the same token, token embedding can vary after aggregation in the middle layers of transformers. If the mask is "token-wise" but not conditioned on the current embedding, how does it adapt to different embeddings for the same token?

    - Does the training need to be conducted for each sparsity ratio?

2. Training Cost of Learnable Masks:

    - What is the computational cost associated with training the learnable masks?

---

### Official Review · Reviewer_L4Xe · 2024-11-08

**Soundness:** 1
**Presentation:** 3
**Contribution:** 2
**Rating:** 5
**Confidence:** 4

**Summary:**

Authors propose MoM, a method that converts a pretrained dense model to a sparsely activated model. MoM constructs experts by dynamically grouping multiple dimensions together in the FFN layers (FFN parameters, hidden states, as well as FFN input) based on the token. During training, sparsity is encouraged by imposing auxiliary loss that aims to obtain a certain hyperparameter of sparsity ratio. All masks are set to 1 (i.e. unmasked) at the start of continue pretraining phase. Authors give downstream evals on 300M models, and training loss comparisons for 8B models. Authors also include various ablation experiments justifying the choice of regularization method and masking strategies.

**Strengths:**

* Efficient training and inference for LLMs is an impactful topic if done correctly.

* The idea of doing a mixture of dynamically determined mask with as an MoE alternative is novel and interesting.

* Description of the method is clear.

* Nice results beating some baselines on the 300M scale.

**Weaknesses:**

See questions below.

**Questions:**

* Equation 2) and 3)  is different from the standard FFN operation used in transformer models, i.e. h=F(W^1 x) W^2 if not using any bias. Why the difference? Authors should instead use the standard FFN formulation for LLMs. Or does the W^g and W^u correspond to the two weight matrices in the SwiGLU activation function? If that’s the case, then 1) the correct notation should have W^g as a input to F, and make it clear F is SwiGLU 2) According to the original FFN output definition (equation 2 in the transformer paper [1].), equation 2 in the paper should also include the last linear layer in the FFN (i.e. the w3 in the llama code base, or “down” in Figure 1).

* Why does MoM not sparsify the w3 (or “down”) parameter in the FFN layers?

* Additional baselines: Can author cite and compare MoM with sparse upcycling [2], which is also a popular approach for turning a single pretrained-dense model to a sparsely activated model. The comparison can be done with matching the number of active parameters as well as training compute with MoM.  For example, take a pre-trained small dense model with N parameters, where N is the number of active parameters in MoM. Then upcycle this small dense model into an MoE and train for a small number of steps.

* A benefit of MoE is that training encourages a balanced router load, thus we can do efficient training and inference with expert parallelization. Does MoM have the same property? Authors should include this comparison in Table 1.

* How does MoM inference cost compare to baselines?

* Is Table 2 for the 300M skywork model only? The biggest concern I have for the paper is that the downstream evals on the 300M scale are not that indicative of model performance  (i.e. it's too small to have above random guessing performance for commonly used benchmarks like GSM8K, MATH, Human Eval, MBPP, ARC Challenge, MMLU etc, which is perhaps why those metrics were not included). And training loss/perplexy results are also not indicative of how good a model is -  can authors include downstream tasks results for Llama 8B or even perhaps larger models on the scale of 2B parameters?


[1] Attention is All You Need, https://arxiv.org/pdf/1706.03762

[2] Sparse Upcycling: Training Mixture-of-Experts from Dense Checkpoints, https://arxiv.org/abs/2212.05055

---

### Note · Authors · 2024-11-24

**Comment:**

We thank the reviewers for their time and valuable feedback. After thorough consideration, we decide to withdraw our paper.

**Withdrawal Confirmation:**

I have read and agree with the venue's withdrawal policy on behalf of myself and my co-authors.